# RETRACTED: Inhibition of EphA2 by Dasatinib Suppresses Radiation-Induced Intestinal Injury

**DOI:** 10.3390/ijms21239096

**Published:** 2020-11-30

**Authors:** Areumnuri Kim, Ki Moon Seong, You Yeon Choi, Sehwan Shim, Sunhoo Park, Seung Sook Lee

**Affiliations:** 1Laboratory of Radiation Exposure and Therapeutics, National Radiation Emergency Medical Center, KIRAMS, Seoul 01812, Korea; ssh3002@kirams.re.kr (S.S.); sunhoo@kirams.re.kr (S.P.); sslee@kirams.re.kr (S.S.L.); 2Laboratory of Biodosimetry, National Radiation Emergency Medical Center, KIRAMS, Seoul 01812, Korea; skmhanul@kirams.re.kr (K.M.S.); c9640@kirmas.re.kr (Y.Y.C.)

**Keywords:** EphA2, dasatinib, intestinal injury, radiation

## Abstract

Radiation-induced multiorgan dysfunction is thought to result primarily from damage to the endothelial system, leading to a systemic inflammatory response that is mediated by the recruitment of leukocytes. The Eph–ephrin signaling pathway in the vascular system participates in various disease developmental processes, including cancer and inflammation. In this study, we demonstrate that radiation exposure increased intestinal inflammation via endothelial dysfunction, caused by the radiation-induced activation of EphA2, an Eph receptor tyrosine kinase, and its ligand ephrinA1. Barrier dysfunction in endothelial and epithelial cells was aggravated by vascular endothelial–cadherin disruption and leukocyte adhesion in radiation-induced inflammation both in vitro and in vivo. Among all Eph receptors and their ligands, EphA2 and ephrinA1 were required for barrier destabilization and leukocyte adhesion. Knockdown of EphA2 in endothelial cells reduced radiation-induced endothelial dysfunction. Furthermore, pharmacological inhibition of EphA2–ephrinA1 by the tyrosine kinase inhibitor dasatinib attenuated the loss of vascular integrity and leukocyte adhesion in vitro. Mice administered dasatinib exhibited resistance to radiation injury characterized by reduced barrier leakage and decreased leukocyte infiltration into the intestine. Taken together, these data suggest that dasatinib therapy represents a potential approach for the protection of radiation-mediated intestinal damage by targeting the EphA2–ephrinA1 complex.

## 1. Introduction

Exposure to high-dose radiation can result in malfunction of the bone marrow, heart, brain, gastrointestinal tract, and lung because these tissues have high concentrations of radiosensitive vascular layers such as endothelial and epithelial cells [1]. Responding to radiation exposure, they produce high levels of pro-inflammatory mediators and reactive oxygen species that induce microvascular leakage, production of high levels of adhesion molecules, and leukocyte infiltration [2]. Previous clinical and experimental studies have shown that radiation-induced vascular damage contributes to systemic tissue injury, thereby accelerating accumulation of lipids, inflammation, and thrombosis [3,4,5].

It is difficult to estimate the expected degree of disease progression and prognosis in the vascular injury by radiation [6]. Thus, to prevent and manage the vascular injury by radiation, early assessment technologies that can physically measure structural changes in blood vessels and blood flow are needed. Recently, various vascular assessments approached predicting the region of vascular disease using nanofluidic devices. These physically analyzed methods can be assessing non-invasive measurement techniques for the promotion of early clinical detection of vascular disease [7,8]. However, due to the lack of data on the underlying molecular and pathological mechanisms of radiation-induced tissue damage, there are no established therapeutic drugs available to treat such injury.

The Eph receptor tyrosine kinase family is composed of nine EphA receptors and five EphB receptors based on sequence homology and preferential binding to their ephrinA and ephrinB ligands [9]. Eph receptors interact with ligands that are often membrane bound, leading to cytoskeletal remodeling, cell motility, and cell adhesion, depending on the levels of Eph and ephrin expression [10]. The interaction of Eph with ephrin results in the activation of various intercellular downstream signaling pathways, including those involving Src family tyrosine kinases, MAPK, p21 kinase, G-proteins, and integrins [11,12]. Most Eph family members have been implicated in tumor progression, as EphA2 is frequently found to be overexpressed in many cancers and enhances cancer proliferation and invasive capacity [13,14]. Moreover, upregulated Eph and ephrin proteins stimulate the production of pro-inflammatory cytokines [15]. Previous studies have shown that monocyte EphB2 receptors bind ephrinB2 on activated endothelial cells, resulting in enhanced monocyte adhesion [16]. This finding suggests that targeting the Eph–ephrin pathway may be a potential therapeutic strategy for various diseases. However, the role of this pathway in radiation-damaged endothelial cells has not been fully elucidated.

Although tyrosine kinase inhibitors are classically used to inhibit tumor cell growth, they can be used to treat other diseases such as pulmonary fibrosis, asthma, and rheumatoid arthritis [17,18,19]. Activated tyrosine kinases play important roles in the regulation of vascular permeability and inflammatory responses of immune-mediated diseases, suggesting that tyrosine kinases are potential therapeutic targets. Dasatinib is a tyrosine kinase inhibitor that blocks various intercellular kinases, including cKit, platelet-derived growth factor receptor, and Btk family members [20]. These kinases and their associated signaling pathways are known to be involved in the recruitment and activation of inflammatory cells. Dasatinib is known to inhibit Src family tyrosine kinases, which are directly involved in the integrin- and Fc-receptor signaling pathways, as well as in the stabilization of neutrophil adhesion [21].

In this study, to analyze vascular injury by radiation, we first measured vessel permeability and rate of infiltrating inflammatory immune cells into the intestine in vivo. Next, we investigated the mechanism by which radiation exposure increases intestinal injury via vascular dysfunction in vitro. Based on the above results, we explored that radiation-activated Eph–ephrin signaling promotes the inflammatory response both in vitro and in vivo. Thus, our findings reveal that treatment with dasatinib attenuates Eph–ephrin signaling in the vascular system, thereby leading to prevention of radiation-induced intestinal damage.

## 2. Results

### 2.1. Radiation Exposure Increases Intestinal Injury In Vivo

Irradiation (IR) increases vascular permeability, which allows leukocyte infiltration into tissues and initiates the inflammatory response [22]. First, we investigated whether IR promotes the destabilization of the intestinal barrier. As shown in Figure 1A, the leakage of Evans blue dye was significantly increased in IR-exposed mice greater than two-fold compared to control mice. Next, to determine whether increased vascular permeability leads to recruitment of immune cells into inflamed tissues, the intestinal lamina propria of irradiated mice were investigated at 3 days after radiation exposure. As shown in Figure 1B, IR recruited inflammatory immune cells, such as myeloid cells (CD11b+), neutrophils (CD11b+Ly6G+), and macrophages (CD11b+F4/80+), into the intestine in a dose-dependent manner. In addition, to confirm the activation of inflammatory immune cell migration by IR, we stained cells in the mouse intestine with the neutrophil marker myeloperoxidase (MPO) and the macrophage marker F4/80. As expected, neither marker was found in unexposed mice; however, these markers were significantly detected in mice 3 days after IR (Figure 1C).

### 2.2. IR Promotes Endothelial Permeability and Mononuclear Cells Adhesion In Vitro

Disruption of endothelial function by radiation is considered one of the hallmarks of the intestinal inflammatory response [23]. We examined the effects on endothelial cell–cell adhesion barrier morphology and function in human umbilical vein endothelial cells (HUVECs) after radiation exposure. As shown in Figure 2A, HUVECs became diffused and swollen after radiation exposure, and the permeability coefficient of FITC–dextran increased by more than five-fold, compared to unexposed cells. In addition, many THP1 cells migrated to the lower chamber of the Transwell in the transmigration assay when co-cultured with HUVECs exposed to IR, whereas very few THP1 cells migrated when co-cultured with non-irradiated HUVECs. As the integrity of the vascular endothelium is largely determined by adherens junctions (AJs), we focused on changes in vascular endothelial (VE)–cadherin, the major AJ molecule. Phosphorylated VE–cadherin has been shown to increase vascular permeability, which downregulates its expression [24]. We found that IR enhanced VE–cadherin phosphorylation on Tyr731. Immunofluorescence staining and Western blotting results show that IR caused a loss of VE–cadherin expression compared to unexposed cells (Figure 2B). We next investigated whether IR could induce the cell adhesion properties of endothelial cells. Human-derived mononuclear cells, such as Jurkat cells and THP1 cells, were co-cultured with HUVECs in Matrigel for 6 h after IR. We observed that cell adhesion was significantly elevated in IR-exposed HUVECs compared to unexposed HUVECs (Figure 2C).

### 2.3. IR Enhances the Expression of Eph Receptors and Their Ligands in Vascular Endothelium

Eph receptors and ephrin ligands are involved in phenotypic changes to the vascular endothelium during conditions of inflammation by the recruitment of fluid and inflammatory cells to inflamed tissues [15]. To determine whether IR could alter the expression of Eph receptors and their ligands in vascular endothelium, the mRNA levels were evaluated in the various irradiated cells inducing leukocytes, T lymphocytes, endothelial and epithelial cells. Among members of the Eph receptor family, EphA2 and EphA3 mRNA levels were significantly increased in response to IR compared to other receptors. EphrinA1 levels were elevated by IR in both Jurkat cells and THP1 cells compared to other ligands (Appendix A). Interestingly, IR did not alter ephrinA/B expression in HUVECs or human intestinal epithelial cells (InEpCs). Eph receptor expression in leukocytes was not significantly altered by IR. Next, to determine whether Eph receptors interact with ephrin ligands under co-culture conditions, IR-exposed HUVECs were either grown alone or in co-culture with Jurkat cells. Radiation exposure increased the phosphorylation of EphA2 in singly cultured HUVECs, and p-EphA2 and p-EphA3 levels were markedly enhanced in co-cultured HUVECs. After radiation exposure, the phosphorylation of ephrinA1 in Jurkat cells was significantly increased in co-cultured HUVECs compared to singly cultured cells (Figure 3A). Similar to the results seen in HUVECs and Jurkat cells, p-EphA2 was elevated after treatment with IR in InEpCs co-cultured with THP1 cells. Higher levels of p-ephrinA1 were also induced by radiation exposure in co-cultured THP1 cells (Figure 3B).

### 2.4. Depletion of EphA2 Inhibits Permeability and Adhesion of Endothelial Cells

We showed that phosphorylation of EphA2 by radiation exposure may be associated with vascular injury. Thus, we silenced EphA2 expression with siRNAs and assessed the functional impact on IR-mediated endothelial barrier disruption and immune cell adhesion. Upon radiation exposure, FITC–dextran leakage was increased in control siRNA-transfected cells, but it was significantly reduced in EphA2-depleted cells (Figure 4A). Next, we assessed the effect of IR on endothelial barrier formation by measuring intercellular VE–cadherin levels. As shown in Figure 4B, IR elevated phosphorylation of VE–cadherin in control siRNA-treated HUVECs but not in EphA2-depleted cells. In addition, we found that leukocyte transmigration was increased when control cells were co-cultured with HUVECs but was decreased in EphA2-depleted cells after radiation exposure. We next evaluated the effects of inhibiting EphA2 in HUVECs on mononuclear cells adhesion. Adhesions of mononuclear cells to monolayers and tube formation of HUVECs were examined when EphA2 was silenced after radiation exposure. In Figure 4C, IR-enhanced adhesion ability of THP1 cells was inhibited in EphA2-depleted HUVECs. We tried to identify the Eph–ephrin signaling involved in IR-induced inflammation in HUVECs. As expected, IR exposure increased the phosphorylation of EphA2, FAK, and the expression of adhesion molecule ICAM1 in HUVECs, whereas these proteins were downregulated in EphA2-depleted cells (Figure 4D).

### 2.5. Dasatinib Reduces IR-Induced Permeability and Adhesion by Inhibiting EphA2

As EphA2 is a receptor tyrosine kinase, we hypothesized that tyrosine kinase inhibitors would suppress endothelial permeability and leukocyte adhesion. To determine whether tyrosine kinase inhibitors could prevent IR-induced endothelial cell damage, we treated several inhibitors for tyrosine kinase to IR-exposed HUVECs, including sorafenib, dasatinib, ibrutinib, and gefitinib. Among them, dasatinib greatly attenuated the IR-increased permeability and VE–cadherin stability compared to untreated cells (Appendix A). Dasatinib also reduced adhesion between endothelial cells and mononuclear cells compared to other inhibitors. Activated EphA2 expression was greatly reduced by dasatinib compared to the other agents. Moreover, dasatinib treatment induced remarkable changes in the in vitro inflammatory system after radiation exposure. It blocked the radiation-activated EphA2 signaling with several proteins in HUVECs and InEpCs, including p-EphA2, p-FAK, ICAM1 and p-VE–cadherin (Figure 5A). Dasatinib treatment inhibited IR-enhanced permeability and adhesion of vascular cells and elevated VE–cadherin disassembly after radiation exposure (Figure 5B). We also found that IR exposure elevated HUVEC–Jurkat cell and InEpC–THP1 cell adhesion, whereas dasatinib significantly suppressed the adhesion rates between these two cell groups (Figure 5C).

### 2.6. Dasatinib Downregulates Phosphorylation of EphA2, Which Prevents Vascular Barrier Disruption and Leukocyte Infiltration into the Intestine

To investigate whether dasatinib suppresses IR-induced intestinal damage in vivo, intestinal morphology of IR-exposed mice was examined after oral treatment of dasatinib for 3 days. IR significantly decreased villus length compared to control mice, whereas dasatinib attenuated IR-induced vascular damage. Dasatinib treatment diminished the crypt length and the thickness of the submucosal layer was greatly enlarged in IR-exposed mice (Figure 6A). The muscularis thickness was not significantly changed. These results show that dasatinib blocked the alteration of intestinal morphology and barrier dysfunction caused by radiation exposure. We also found that increased phosphorylation of EphA2 in Ser897 and Tyr722 in IR-exposed mice was suppressed by dasatinib treatment (Figure 6B). Dasatinib treatment significantly suppressed the hyperpermeability of blood vessels induced by IR (Figure 6C). CD68, a marker of inflammation and the systemic immune response, was expressed in monocyte lineage cells. After radiation exposure, recruitment of CD68-positive inflammatory immune cells was detected in the intestine, but dasatinib attenuated the infiltration of these cells into the intestine (Figure 6D). In addition, dasatinib blocked the infiltration of IR-induced inflammatory myeloid cells, including CD11b+, CD11b+Ly6G+, and CD11b+F4/80+, into the intestinal lamina propria (Figure 6E).

## 3. Discussion

Radiation is widely used to diagnose and treat certain medical conditions. Despite this benefit, radiation at high doses (>0.5 Gy) can induce toxicity in normal tissues. In patients with pelvic cancer who received radiotherapy for 5 to 6 weeks, approximately 80% will develop gastrointestinal impairments [25]. As the gastrointestinal track is composed of microvascular endothelial and mucosal epithelial cells [1], IR exposure could cause normal cell damage and changes in immune system function in intestinal tissue. Thus, vascular dysfunction is considered one of the pivotal factors of inflammatory intestinal disease.

Vascular endothelium dysfunction is a systemic disorder characterized by vascular hyperpermeability, increased blood flow and recruitment of inflammatory cells, which initiates process of atherosclerosis and directly relates to numerous vascular diseases [26]. Therefore, the precise measurement of altered blood flow and vascular structure is important to reduce vascular damage [27]. Nowadays, computational fluid dynamics techniques are widely applied for early assessment of vascular disease using three dimensional models [28]. These techniques combined with clinical assessments could approach the vascular health status and precious information regarding arteries [29,30]. Thus, we expected that the use of physiologically-based techniques would accurately predict the risk of IR-induced vascular disease. Although some biological studies have suggested that dysregulated vascular function leads to systemic inflammatory response syndrome, the associated mechanism of radiation-induced intestinal inflammation has not been elucidated [31,32].

In the current study, we showed that IR exposure resulted in intestinal injury via the loss of vascular integrity and infiltration of inflammatory immune cells into the intestine. We also demonstrated that increasing the expression of EphA2 and EphrinA1 by IR promoted breakdown of vascular permeability and immune cells adhesion. Moreover, we found that the inhibition of EphA2 by dasatinib reduced IR-induced vascular damage, suggesting that dasatinib could be an attractive anti-inflammatory therapy for patients with radiation exposure.

VE–cadherin is an AJ complex, which plays an important role in the maintenance of vascular integrity. Phosphorylation of VE–cadherin on Tyr685 results in its destabilization, which permits vascular barrier leakiness and transendothelial migration of leukocytes [33]. In addition, the phosphorylation of FAK on Tyr397 allows its interaction with VE–cadherin to disrupt the AJ and alter endothelial permeability [34]. Consistent with this finding, we observed that IR increased endothelial VE–cadherin phosphorylation on Tyr685 and accompanied by FAK phosphorylation, leads to endothelial hyperpermeability in vitro (Figure 2 and Figure 4). Moreover, depletion of EphA2 using siRNAs revealed that the disruption of VE–cadherin and other endothelial permeability mediators by FAK activation in IR-exposed HUVECs was blocked. These data propose that endothelial EphA2 activation leads to the rearrangement of VE–cadherin and cytoskeletal components by FAK, resulting in increased endothelial barrier destabilization. We also found that endothelial/epithelial cells cultured with mononuclear cells resulted in increased phosphorylation of EphA2 compared to mono-cultured cells after radiation exposure, suggesting that elevated EphA2 plays an important role in inflammatory response (Figure 3). Previous studies have suggested that Eph receptors and Ephrin ligands could be involved in the inflammatory response but may act differently depending on the type of stimulus and tissue [35,36,37]. Accordingly, our results indicate that the upregulation of EphA2–ephrinA1 promotes the destabilization of the endothelial–epithelial barrier and immune cell adhesion, suggesting that the inhibition of EphA2 and ephrinA1 could serve as a potential treatment for intestinal inflammation induced by IR.

Dasatinib, a multiple tyrosine kinases inhibitor, was found to effectively suppress the radiation-induced vascular dysfunction (Figure 5 and Figure 6). In vascular endothelium such as endothelial and epithelial cells, it markedly inhibited the phosphorylation of EphA2 and FAK, accompanied by the reduced expression of ICAM1 and p-VE–cadherin. These data propose that dasatinib treatment is a potential strategy to ameliorate the radiation-induced vascular dysfunction and inhibit EphA2 activation. Moreover, this is also supported by the experimental evidence that EphA2 mRNA is highly expressed upon radiation exposure among the Eph receptors (Appendix A). Radiation-induced mononuclear cell adhesion, such as THP1 and Jurkat cells, was markedly decreased by dasatinib treatment (Figure 5). Dasatinib inhibits Bcr-Abl and Src family members by targeting threonine gatekeeper residues [38]. Most Eph receptors also contain a threonine gatekeeper residue [39]. Thus, we hypothesized that dasatinib would effectively block radiation-induced vascular destabilization by potently inhibiting the threonine gatekeeper residue in EphA2.

Interestingly, it suppressed the activated Src family kinase by radiation exposure without reduction in phosphorylated ephrinA1, the ligand for EphA2 receptor (Appendix A). Similar to our findings, dasatinib has been reported to potentially contribute to the treatment of chronic inflammatory disease by blocking the Src family kinases involved in neutrophil adhesion [21]. Src family kinases have been shown to be essential for the recruitment and activation of immune cells; Src has also been suggested to promote Rho activation by FAK following the activation of EphA2, leading to vascular permeability and the inflammatory response [40].

Collectively, these observations indicate that in cases of acute radiation syndrome, the EphA2–ephrinA1 signaling pathway facilitates vascular permeability and mononuclear cell–endothelial cell adhesion, resulting in damage to intestinal tissues. Inhibition of the Eph–ephrin signaling pathway with dasatinib appears to be a promising strategy for the treatment of patients with gastrointestinal damage caused by IR exposure.

## 4. Materials and Methods

### 4.1. Cell Culture and Reagents

The human umbilical vein endothelial cell (HUVEC) line was purchased from Lonza Biologics Inc. (Portsmouth, NH, USA) and cultured in EGM^TM^-2 (Lonza) supplemented with 5% fetal bovine serum (FBS). Cells were passaged four to six times before use in this study. Human intestinal epithelial cells (InEpCs) were obtained from Lonza and cultured in SmGM^TM^-2 containing supplements and 5% FBS. The THP1 monocyte cell line and T lymphoblast Jurkat cells were obtained from the American Type Culture Collection (ATCC, Manassas, VA, USA). These cells were cultured in RPMI1640 containing 10% FBS. All cells were cultured in 5% CO_2_ at 37 °C.

### 4.2. In Vivo Radiation Exposure and Dasatinib Treatment

Seven week-old male C57BL/6 mice were maintained under specific pathogen-free conditions and acclimated for at least 7 days before handling. The animals were exposed to whole-body irradiation (IR) using an X-ray machine (X-RAD 320, North Branford, CT, USA) at a dose rate of 1 Gy/min. Dasatinib was dissolved in equal parts propylene glycol and water and orally administered (1 mg/kg/day). To determine the effects of dasatinib on mouse vessel permeability and leukocyte infiltration, mice were irradiated with 5 Gy, and dasatinib was injected daily for 3 days after IR. All mouse experiments were performed in accordance with the Korea Institute of Radiological and Medical Science Institutional Animal Care and Use Committee (IACUC)-approved protocol (No. Kirams 2017-0010, 23 February 2017)

### 4.3. Transwell Migration Assay

Transwell assays were performed using Transwell 24-well plates (Costar, Corning Inc., Corning, NY, USA). To assess the rate of cell migration, a total of 5 × 10^4^ HUVECs were added to the upper chamber of the plate after 12 h to form a monolayer. Cells were irradiated for 24 h. A total of 1 × 10^4^ carboxyfluorescein succinimidyl ester (CFSE)-labeled THP1 cells were starved for 3 h and added to the upper chamber of each Transwell for co-culture with HUVECs. RPMI1640 media containing 0.1% BSA was added to the lower chamber. Transmigration was allowed to occur for 2 h at 37 °C. Transmigrated THP1 cells were quantified. Photographs of leukocyte adhesion were obtained using confocal microscopy (Leica, Wetzlar, Germany).

### 4.4. Adhesion of Leukocytes to Endothelial Tubes

To build tube formation of endothelial cells, a total of 2 × 10^4^ PKH26-labeled HUVECs or InEpCs were seeded onto a Matrigel-coated 96-well plate. On the following day, HUVECs and InEpCs were irradiated and then 1 × 10^4^ CFSE-labeled Jurkat cells or THP1 cells were added to the plate and co-cultured with the HUVECs at 37 °C. After 2 h, the cells were washed with PBS, and adherent cells were quantified.

### 4.5. In Vitro Permeability Assay

The permeability of the HUVEC monolayer was determined using an in vitro vascular permeability assay (Merck Millipore, Burlington, MA, USA) and performed according to the manufacturer’s instructions. Briefly, 2 × 10^4^ HUVECs were seeded in a 24-well plate and incubated overnight. On the following day, the cells were irradiated for 24 h; 20 μg/mL FITC–dextran was then added to the upper chamber. After incubation at 37 °C for 30 min, 100 μL of the medium was drawn from the lower chamber and examined using a microplate reader. The fluorescence intensity of the medium was measured in duplicate per condition and normalized to untreated control cells.

### 4.6. In Vivo Permeability Assay

Mice were subjected to IR for 3 days. Evans blue (0.6 mg/g body weight, 4 kDa, Sigma-Aldrich, St. Louis, MO, USA) was then injected via the tail vein, and mice were sacrificed 4 h later. To determine the concentration of Evans blue in the tissues, 100 mg of intestinal tissue was incubated in 500 μL formamide at 56 °C for 48 h, and the optical density of the solution at 600 nm was compared to a standard curve.

### 4.7. Isolation of Leukocytes from the Lamina Propria and FACS Analysis

Leukocytes were isolated from the lamina propria by enzymatic tissue digestion, according to a modified method of Goodyear et al. [41]. Briefly, after depletion of the epithelial layer, the mucosal tissue was cut into 2–4-mm pieces and digested in a shaking (200–300 rpm) water bath at 37 °C for 1.5 h with collagenase IV (70 mg/mL; Sigma-Aldrich), EDTA (5 mM), and DTT (0.145 mg/mL; Sigma-Aldrich) in serum-free RPMI1640 medium containing 2% L-glutamine. The tissue fragments were then digested with shaking (at ~400 rpm) in pre-warmed serum-free RPMI1640 medium containing Liberase (0.25 mg/mL) and 0.05% DNase for 30 min at 37 °C in 5% CO_2_. The digested tissue was collected in RPMI1640 medium containing 3% FBS and filtered through a 70 μm cell strainer. The cell suspension was centrifuged for 10 min at 1350 rpm at 4 °C, resuspended in 10 mL serum-free media, and filtered through a 40 µm cell strainer on ice. For cell surface receptor expression analysis, both non-irradiated and irradiated cells were kept on ice, washed with PBS, and gently resuspended in FACS buffer (PBS, 2% FBS, and 0.01% sodium azide). Cells were then incubated with anti-mouse CD45-APC, CD11b-FITC, Gr1-PE, Ly6G-PE, and F4/80-PE/Cy7 (BioLegend, San Diego, CA, USA) antibodies in the dark for 30 min. Isotype-matched antibodies were used as negative controls to define background staining. Images were acquired using the FACSCanto analyzer (BD Biosciences, San Jose, CA, USA) and Cell Quest software. Analyses were then performed using FlowJo software (v7.6.5).

### 4.8. Immunohistochemistry

Immunohistochemical staining was performed as described previously [42]. To detect neutrophil and macrophage infiltration after IR, sections of mouse intestine were stained with MPO and F4/80 (Abcam, Cambridge, UK). To determine the effects of IR and dasatinib treatment, sections of mouse intestine were stained with anti-phospho-EphA2 (Ser897) (Cell Signaling Technology, Danvers, MA, USA). Histologic staining was performed with hematoxylin and eosin. To stain for CD68, a monocyte lineage marker, irradiated and dasatinib-treated mouse intestine was fixed in 20% sucrose overnight and embedded in optimal cutting temperature (OCT) compound (Thermo Fisher Scientific, Waltham, MA, USA). Cryosections (5–10 μm) were incubated with primary antibody against CD68 for 3 h and then with secondary anti-rat-FITC antibody for 1 h. Immunohistochemically stained slides were examined using confocal laser scanning microscopy (Leica).

### 4.9. Immunofluorescence Staining

Samples were prepared and stained as described previously [42]. Cells were exposed to radiation (5 Gy) for 24 h. Primary antibodies specific for VE–cadherin (Abcam) were used, and DAPI (Sigma-Aldrich) was used for nuclear staining. Immunostained slides were observed using confocal laser scanning microscopy.

### 4.10. Real-Time PCR

Total cellular RNA was isolated using the TRIzol reagent (Invitrogen, Carlsbad, CA, USA) and subjected to reverse transcription to obtain total cDNA using Power SYBR Green Master Mix-based qRT PCR (Thermo Fisher Scientific). Real-time RT-PCR was performed using a LightCycler 489 Real-Time PCR System (Roche, Indianapolis, IN, USA) and the standard settings. Target gene expression was normalized to levels of GAPDH. The sequences of the primers used are shown in Appendix A.

### 4.11. Western Blotting and Immunoprecipitation Analysis

Western blotting was performed as described previously [43] using primary antibodies against the following proteins: p-EphA2 (Tyr722 and Ser897), EphA2, p-EphA3 (Tyr779), EphA3, p-ephrinA1, ephrinA1, p-NF-κB, p-Tyr, p-focal adhesion kinase (FAK), p-Src, Src, p-Syk and Syk (Cell Signaling Technology); p-VE–cadherin and VE–cadherin (Abcam), and ICAM1 and β-actin (Santa Cruz Biotechnology, Dallas, TX, USA). Immunoprecipitation assays were performed as described previously [42]; briefly, cell lysates were incubated with anti-p-Tyr antibody and protein A-Sepharose beads (GE Healthcare, Pittsburgh, PA, USA) overnight at 4 °C. Immunoprecipitates were collected and subjected to Western blot analysis.

### 4.12. siRNA Transfection Assay

HUVECs (1 × 10^5^ cells/well) were seeded into six-well plates and transfected with 30 nM EphA2 siRNA (Dharmacon, Lafayette, CO, USA) or control non-targeting siRNA (Dharmacon) using Lipofectamine 2000, according to the manufacturer’s protocol. After 12 h, cells were either irradiated or not for 24 h.

### 4.13. Statistical Analyses

Quantitative data are expressed as the mean ± standard deviation (SD). Mean values were compared using a one-way analysis of variance and the Student’s t test. A *p* value less than 0.05 was considered statistically significant. All experiments were conducted in triplicate.

## 5. Conclusions

We demonstrated that radiation exposure caused damage of vascular endothelium through activated EphA2–ephrinA1 signaling, resulting in intestinal inflammation. The determined results are as follows: first, the vascular permeability, infiltrating immune cells and cell–cell adhesion by radiation was increased about three-fold compared to the control. Second, radiation exposure increased interaction between endothelium EphA2 and immune cells EphrinA1, thereby promoting vascular damage and inflammatory response in vitro and in vivo. Third, treatment with dasatinib ameliorates the radiation-induced intestinal injury similar to that of the control through halted EphA2 phosphorylation, and inhibited infiltration of inflammatory cells into the intestinal tissue. Therefore, this study suggests that treatment with dasatinib could be potential therapeutic approach for decreasing radiation-induced intestinal injury.

## Figures and Tables

**Figure 1 ijms-21-09096-f001:** Irradiation (IR) increases intestinal permeability and leukocyte infiltration. (**A**) Mice were subjected to whole body IR (5 Gy). After 3 days, mice were injected with Evans blue solution (100 µL of 30 mg/mL) via the tail vein and sacrificed after 30 min. The intestines were collected and photographed. The amount of extracted Evans blue dye was quantified. Results were normalized by tissue weight (mean ± SD); n = 8 control and 8 irradiated mice, * *p* < 0.01. (**B**) Three days following IR, intestinal lamina propria cells were isolated and stained for myeloid cells (CD11b+), neutrophils (CD11b+Ly6G+), and macrophages (CD11b+F4/80+) and analyzed using flow cytometry. Graphs show the percent of the CD45+ cell population (mean ± SD); n = 5. * *p* < 0.01. (**C**) Representative results of the immunohistochemical staining of control and IR-treated intestines for myeloperoxidase (MPO) and F4/80 3 days after IR (7 Gy). Arrows, stained cells. Graphs show the MPO or F4/80 positive cell population (mean ± SD); n = 5. * *p* < 0.01.

**Figure 2 ijms-21-09096-f002:** IR enhances vascular barrier permeability and leukocyte adhesion. (**A**) Representative images show the morphology of human umbilical vein endothelial cells (HUVECs) and transmigration of THP1 cells co-cultured with HUVECs 24 h after IR. The graph shows the amount of FITC–dextran leakage. Results were normalized to those in control cells (mean ± SD); * *p* < 0.01. (**B**) Twenty-four hours after IR, immunofluorescence staining for vascular endothelial (VE)–cadherin in IR-treated HUVECs was performed. VE–cadherin and phosphorylated VE–cadherin levels were measured by Western blotting. β-actin and p-tyr were used as protein loading controls. (**C**) HUVECs were seeded onto Matrigel, subjected to IR (5 Gy), and co-cultured with leukocytes (THP1 and Jurkat cells) for 6 h. The graph shows the relative adhesion rate of leukocytes to HUVECs (mean ± SD); * *p* < 0.01.

**Figure 3 ijms-21-09096-f003:** Phosphorylated EphA2 and ephrinA1 expression levels are upregulated by radiation exposure. Twenty-four hours after IR, p-EphA2, EphA2, p-EphA3, EphA3, p-ephrinA1, and ephrinA1 expression levels were measured in whole-cell lysates by Western blotting. (**A**) HUVECs were exposed to IR and either co-cultured with Jurkat cells (co) or singly cultured (monoculture; mono). (**B**) Human intestinal epithelial cells (InEpCs) were exposed to IR and either co-cultured with THP1 cells (co) or singly cultured (monoculture; mono). *β*-actin was used as the protein loading control. Bars represent mean ± SD of 3 independent experiments. * *p* < 0.01.

**Figure 4 ijms-21-09096-f004:** Depletion of EphA2 with siRNAs blocks IR-induced endothelial cell damage. (**A**) HUVECs were transfected for 24 h with EphA2 siRNAs, which was followed by IR (5 Gy) exposure for an additional 24 h. Endothelial permeability is represented by the amount of FITC–dextran staining. * *p* < 0.05. (**B**) Twenty-four hours after IR, siCont or siEphA2-treated HUVECs were stained with FITC-conjugated p-VE–cadherin and visualized using immunofluorescence (upper panel). Carboxyfluorescein succinimidyl ester (CFSE)-labeled THP1 cell transmigration when co-cultured with siCont or siEphA2-treated HUVECs (lower panel). Western blotting results show levels of p-VE–cadherin in siCont or siEphA2-treated HUVECs after IR. P-tyr was used as a protein loading control. (**C**) Monolayers or tubes of PKH26-labeled HUVECs (red) were exposed to IR and then co-cultured with CFSE-labeled THP1 cells (green) for 24 h. The graph shows the relative adhesion rate. * *p* < 0.05. (**D**) Expression levels of p-EphA2, EphA2, ICAM1, and p-FAK in HUVECs were determined using Western blotting. *β*-actin was used as a protein loading control.

**Figure 5 ijms-21-09096-f005:** Downregulation of EphA2 by dasatinib reduces IR-induced permeability and adhesion. (**A**) HUVECs and InEpCs were treated with dasatinib (500 nM) and then exposed to IR for 24 h. Expression levels of p-EphA2 (Ser897 and Tyr588), EphA2, p-FAK, ICAM1 and p-VE–cadherin were measured by Western blotting (left panel). (**B**) Twenty-four hours after IR (5 Gy), VE–cadherin expression and transmigration were analyzed in HUVECs with or without dasatinib (500 nM). HUVEC permeability is represented by the amount of FITC–dextran staining. * *p* < 0.05. (**C**) PKH26-labeled HUVECs (red) were exposed to IR (5 Gy) with or without dasatinib (500 nM) for 12 h, followed by co-culture with CFSE-labeled Jurkat cells (green). PKH26-labeled InEpCs (red) were exposed to IR with or without dasatinib for 12 h, followed by co-culture with CFSE-labeled THP1 cells (green). The adhesion rate was measured at 6 h. * *p* < 0.05.

**Figure 6 ijms-21-09096-f006:** Dasatinib suppresses intestinal injury by radiation exposure in mice. Mice were exposed to IR (5 Gy) followed by treatment with dasatinib. Twenty-seven days after IR, mice were sacrificed. (**A**) Representative images show hematoxylin and eosin (H&E) staining from intestine of control, IR, dasatinib (DST) and IR+DST animals. The sections were analyzed for villus length (1), crypt depth (2), submucosa thickness (3) and muscularis thickness (4) (original magnification ×40, insert ×100). Each bar represents mean ±SD of 7–9 analyzed sections per animal (n = 5). * *p* < 0.05. (**B**) Representative images show EphA2 (Ser897) expression in mouse intestine using immunohistochemical staining. The graph indicates the percentage of p-EphA2-positive cells. Expression of p-EphA2 (Ser897 and Tyr588) was measured in mouse intestinal tissues by Western blotting; n = 4. *β*-actin was used as a protein loading control.; mean ± SD, * *p* < 0.05. (**C**) Intestinal permeability was assessed using the Evans blue assay. Representative images show mouse intestine. The graph shows the measurement of Evans blue dye leakage into the intestine; mean ± SD, * *p* < 0.05. (**D**) Immunofluorescence staining of CD68 in mouse intestine. The graph shows the percentage of CD68-positive cells; mean ± SD, * *p* < 0.05. (**E**) Mouse intestinal lamina propria cells were stained for CD45/CD11b/Ly6G/F480 and analyzed by flow cytometry. Representative images show CD11b+ (myeloid cells), CD11b+Ly6G+ (neutrophils), and CD11b+F4/80+ (macrophages); mean ± SD, n = 5, * *p* < 0.05.

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
