# Peer review of "Inhibition of EphA2 by Dasatinib Suppresses Radiation-Induced Intestinal Injury"

_ijms, 2020, doi:10.3390/ijms21239096_

Round 1

Reviewer 1 Report

  • I recommend this paper be published after major revisions. There are some suggestions for the authors.
  • Format of the reference list should be checked and prepared according to the journal's rule
  • The language is fluent overall, needing some improvements and typos should be removed in some places.
  • What is the main practical contribution of your work? elaborate in the concussion.
  •  
  • More relevant references should be added. Authors can read the article
  • https://doi.org/10.1016/j.physa.2020.124669
  •  
  • Please, ask a native English speaker for the improvement of the manuscript.
  • The English quality of the manuscript must be improved and the equations better explained, with all the symbols and subscripts clearly defined, before the manuscript  can be considered for evaluation.
  • The current literature review is not sufficient. For example, please mention (give) following more current study in the sections of Introduction, and References List for completeness of your study and the references:
  • The logic of the introduction is not clear enough so that the reviewer is confused to some degree. In order to descript the process and significance of present object clearly, please clarify the technical logic of present object to improve the introduction.
  • The results of the work should be compared with the results of other investigators and/or other methodologies, experimental results or even simulated results. This is needed to place this work in perspective with other work in the field and provide more credibility for the present results.
  • To enrich the introduction section, the following published paper can be cited:
    • https://doi.org/10.1016/j.cmpb.2019.105170
  • The conclusion section is not proper and has to be totally revised. This section should clearly demonstrate the results by percentage and compare them by data and numbers to show the best scenario.
  •  

Author Response

I recommend this paper be published after major revisions. There are some suggestions for the authors.

  1. Format of the reference list should be checked and prepared according to the journal's rule

Answer: Following referee pointed out, we have revised the format of the reference list as well as the whole manuscript according to the style of ‘International Journal of Molecular Sciences’.

  1. The language is fluent overall, needing some improvements and typos should be removed in some places.

Answer: We apologize for the mistake. We totally examined the manuscript and changed some typo-errors as the reviewer pointed out. We have attached the certificate of English editing service (no. 39157 in Bioscience Writers)

  1. What is the main practical contribution of your work? Elaborate in the concussion.

Answer: Following the reviewer’s opinion, we described the value of this study at conclusion section on page 14, lines 454-463. Many previous studies have shown that radiation exposure increased normal cell or tissue damage, leading to systemic inflammation and vascular diseases. However, radiation-associated molecular mechanism was not fully understood until now. We observed that radiation increased activation of endothelial/epithelial EphA2 and immune cells EphrinA1, which aggravates vascular permeability and cell-cell adhesion. In addition, we showed that pharmacological inhibition of EphA2-EphrinA1 by the dasatinib attenuated the inflammatory response in vitro and in vivo. Therefore, this study suggests that targeting EphA2-EphrinA1 signaling by dasatinib could be potential drug for patients with radiation-induced intestinal injury.

  1. More relevant references should be added. Authors can read the article https://doi.org/10.1016/j.physa.2020.124669.

Answer: As refree’s comments, we added the references to introduction section with a relevant context. Please check the main body of introduction part on page 1, line 42-47.

  1. Please, ask a native English speaker for the improvement of the manuscript. The English quality of the manuscript must be improved and the equations better explained, with all the symbols and subscripts clearly defined, before the manuscript can be considered for evaluation.

Answer: This manuscript was edited by English-editing service, BioScience Writers (Huston, TX). Please see the attached invoice (No.39157), and totally revised again. We agree to the reviewer’s opinion that numerical investigation with equation is very efficient way to explain the experimental results, including the accurate measurement for the radiation-induced vascular damage. In this study, we have used indirect methods to analyze vascular permeability such as FITC-dextran permeability assay (in vitro) and leakage of Evans blue (in vivo). In addition, we have measured the infiltration rate of inflammatory immune cells into intestine to determine whether radiation increase cell-cell adhesion. These methods are conventionally used to measure the inflammatory response in the biological studies and they could be conceived as the general experimental methods without specific numeric equation. Instead, we numerically measured the change of analyzed results and expressed as graphical data, which could not be expressed with an equation due to some uncontrollable factor in physiological change. Considering the importance of computational investigation on the physiological study, we have mentioned them in the main text of manuscript introduction section page on 1, lines 41-49 and discussion section on page 10, lines 270-278. Here, we have attached recent references regarding on the related research.

  • Human Intestinal Mononuclear Phagocytes in Health and Inflammatory Bowel Disease, Front Immunol. 2020; 11: 410.
  • ADAM10 regulates endothelial permeability and T-Cell transmigration by proteolysis of vascular endothelial cadherin, Circ Res. 2008 May 23;102:1192-201.

  1. The current literature review is not sufficient. For example, please mention (give) following more current study in the sections of Introduction, and References List for completeness of your study and the references:

Answer: Following the reviewer’s opinion, the relevant paragraph in the introduction was totally revised and the added new relevant references to complement the contents. Please check the text in the introduction section.

  1. The logic of the introduction is not clear enough so that the reviewer is confused to some degree. In order to descript the process and significance of present object clearly, please clarify the technical logic of present object to improve the introduction.

Answer: As refree’s pointed out, we have revised the manuscript more clearly and mentioned the technical procedure of experiment in the introduction part page on 2, line 74~80.

  1. The results of the work should be compared with the results of other investigators and/or other methodologies, experimental results or even simulated results. This is needed to place this work in perspective with other work in the field and provide more credibility for the present results. To enrich the introduction section, the following published paper can be cited: https://doi.org/10.1016/j.cmpb.2019.105170

Answer: As you commented, data comparison analysis is helpful to understand the experimental findings, which can strengthen the value of scientific evidence. We described comparative explanation of our data with other methodological studies in the discussion together with attached citation of references. Especially, the paper you commented is a very useful example for the enhancement of value of our experimental findings through the comparison. We have mentioned them in the main text of manuscript discussion section page on 10, lines 270-278.

  1. The conclusion section is not proper and has to be totally revised. This section should clearly demonstrate the results by percentage and compare them by data and numbers to show the best scenario.

Answer: We appreciate the sincere comments. As the reviewer pointed out, we’ve revised the conclusion part page on 14, lines 454-463.

Reviewer 2 Report

Areumnuri Kim et al. present new data about the Ephrin-mediated mechanisms of radiation-induced intestinal injury and the positive effects of dasatinib in radiation-induced intestinal injury. The role of Ephrins in the head-and-neck and non-small cell lung cancers was documented before, and the role of Ephrin B signaling in wound-healing and regeneration of the intestinal epithelium has also been described. The study of Kim et al. is a well-designed, innovative experimental work, with original results on the role of Ephrin/Ephrin receptor signaling. The manuscript is well-structured and clearly illustrated, and the conclusions are in line with the authors' findings. The main findings presented are:

  1. Irradiation causes a dose-dependent increase of intestinal permeability and CD11b+Ly6G+ neutrophil and CD11b+F4/80 macrophage infiltration.
  2. IR enhances endothelial permeability with FITC-dextran leakage through VE-cadherin phosphorylation on Tyr731. Increased adhesion of Jurkat and THP1 cells co-cultured with HUVEC cells.
  3. Radiation exposure did not affect ephrin ligands in HUVECS; instead, it increased the phosphorylation of EphA2 in single-cultured HUVECs, and phosphorylated EphA2 and EphA3were significantly increased, and p-EphA3 levels were also up-regulated in co-cultured HUVECs. 
  4. Radiation triggered the phosphorylation of ephrinA1 in Jurkat cells co-cultured with HUVECs, and intestinal epithelial cells showed high p-EphA2 levels after IR treatment with IR when co-cultured with THP1 cells.
  5. Leukocyte transmigration and the expression of ICAM-1 were decreased in the case of EphA2 knock-down by siRNA.
  6. Out of several tyrosine kinase inhibitors, dasatinib reduced EphA2 expression and suppressed cell adhesion.
  7. Dasatinib treatment causes morphological changes in the intestinal mucosa, halted EphA2 phosphorylation, and inhibited inflammatory cells' infiltration into the intestinal tissue.

It is important to underline this research's scientific value, which obviously qualifies it for publication in the International Journal of Molecular Sciences. I have only a few small remarks to improve further the manuscript's quality:

  1. I suggest using the definition "mononuclear cells" instead of leukocytes in section Results 2.2 and Figure 2. Jurkat cells are lymphocytes, whereas THP1 is a monocyte cell line.
  2. The authors mention that they performed experiments also with other tyrosine-kinase inhibitors: sorafenib, ibrutinib, and gefitinib. Were these results divergent from those with dasatinib? If so, what are the explanations? It would be worthwhile to mention synthetically, in several phrases, also these observations.
  3. A list of abbreviations would be useful for an easier understanding of the text. 
  4. The English of the manuscript is fine; however, in some places, small polishing would be beneficial:

e.g., 2.3. IR enhances the expression of Eph receptors and their ligands in vascular system

In conclusion, I evidently recommend the valuable manuscript of Kim et al. to be published in the International Journal of Molecular Sciences.

Author Response

Reviewer 2

Areumnuri Kim et al. present new data about the Ephrin-mediated mechanisms of radiation-induced intestinal injury and the positive effects of dasatinib in radiation-induced intestinal injury. The role of Ephrins in the head-and-neck and non-small cell lung cancers was documented before, and the role of Ephrin B signaling in wound-healing and regeneration of the intestinal epithelium has also been described. The study of Kim et al. is a well-designed, innovative experimental work, with original results on the role of Ephrin/Ephrin receptor signaling. The manuscript is well-structured and clearly illustrated, and the conclusions are in line with the authors' findings. The main findings presented are:

  1. Irradiation causes a dose-dependent increase of intestinal permeability and CD11b+Ly6G+ neutrophil and CD11b+F4/80 macrophage infiltration.
  2. IR enhances endothelial permeability with FITC-dextran leakage through VE-cadherin phosphorylation on Tyr731. Increased adhesion of Jurkat and THP1 cells co-cultured with HUVEC cells.
  3. Radiation exposure did not affect ephrin ligands in HUVECS; instead, it increased the phosphorylation of EphA2 in single-cultured HUVECs, and phosphorylated EphA2 and EphA3were significantly increased, and p-EphA3 levels were also up-regulated in co-cultured HUVECs. 
  4. Radiation triggered the phosphorylation of ephrinA1 in Jurkat cells co-cultured with HUVECs, and intestinal epithelial cells showed high p-EphA2 levels after IR treatment with IR when co-cultured with THP1 cells.
  5. Leukocyte transmigration and the expression of ICAM-1 were decreased in the case of EphA2 knock-down by siRNA.
  6. Out of several tyrosine kinase inhibitors, dasatinib reduced EphA2 expression and suppressed cell adhesion.
  7. Dasatinib treatment causes morphological changes in the intestinal mucosa, halted EphA2 phosphorylation, and inhibited inflammatory cells' infiltration into the intestinal tissue.

 It is important to underline this research's scientific value, which obviously qualifies it for publication in the International Journal of Molecular Sciences. I have only a few small remarks to improve further the manuscript's quality:

  1. I suggest using the definition "mononuclear cells" instead of leukocytes in section Results 2.2 and Figure 2. Jurkat cells are lymphocytes, whereas THP1 is a monocyte cell line.

Answer: As the reviewer pointed out, we changed the ‘leukocytes’ with ‘mononuclear cells’ in whole manuscript.

  1. The authors mention that they performed experiments also with other tyrosine-kinase inhibitors: sorafenib, ibrutinib, and gefitinib. Were these results divergent from those with dasatinib? If so, what are the explanations? It would be worthwhile to mention synthetically, in several phrases, also these observations.

Answer: We think the reviewer’s point is reasonable. We described them in discussion section page on 10, lines 314-318.

“Dasatinib inhibits Bcr-Abl and Src family members by targeting threonine gatekeeper residues [38]. Most Eph receptors also contain a threonine gatekeeper residue [39]. Thus, we hypothesized that dasatinib would effectively block radiation-induced vascular destabilization by potently inhibiting the threonine gatekeeper residue in EphA2.”

  1. A list of abbreviations would be useful for an easier understanding of the text. 

Answer: As refree’s pointed out, we have added a list of abbreviations for an easier understanding of the text.

  1. The English of the manuscript is fine; however, in some places, small polishing would be beneficial:e.g., 2.3. IR enhances the expression of Eph receptors and their ligandsin vascular system

Answer: As refree’s comment, we revised manuscript polishing the rough expression, including ‘vascular system’ to ‘vascular endothelium’. 

“2.3. IR enhances the expression of Eph receptors and their ligands in vascular endothelium.”

  1. In conclusion, I evidently recommend the valuable manuscript of Kim et al. to be published in the International Journal of Molecular Sciences.

Answer: We deeply appreciate your review of this work. We are hopeful that this revision will satisfy the criteria for publication of the Editorial board of the journal.
